# Effect of Beef Silver Skin (Epimysium) Levels on Meat Emulsion Stability, Quality Attributes, and Texture Parameters

**DOI:** 10.3390/foods12203775

**Published:** 2023-10-14

**Authors:** Kentaro Kawata, Francine M. Giotto, Amilton S. de Mello, Thomas Kingery, Luiz H. P. Silva

**Affiliations:** 1Carol Martin Gatton Academy of Mathematics and Science, Western Kentucky University, Bowling Green, KY 42101, USA; kkawata@umass.edu; 2Department of Agriculture, Nutrition and Veterinary Sciences, University of Nevada Reno, Reno, NV 89557, USA; fgiotto@nmsu.edu (F.M.G.); ademello@unr.edu (A.S.d.M.); 3Department of Animal and Range Sciences, New Mexico State University, Las Cruces, NM 88003, USA; 4Department of Agriculture and Food Science, Western Kentucky University, Bowling Green, KY 42101, USA; thomas.kingery@wku.edu

**Keywords:** collagen, emulsion stability, fat loss, silver skin, texture profile

## Abstract

The epimysium, also known as silver skin, is a fascia of connective tissue that surrounds each muscle. During fabrication, epimysium is removed from intact cuts, and it can be used as a source of collagen in processed meats to reduce production costs. However, little is known about the emulsifying properties of this collagen source. Thus, this study aimed to evaluate the effect of three levels of beef epimysium (silver skin, 0, 5, and 10%) on meat emulsion stability and on its cooked characteristics. Beef silver skin partially replaced ground beef, pork, and fat trimming, while all the other ingredients remained constant across formulations. The inclusion of silver skin did not affect (*p* > 0.05) chemical composition, total cooking loss, water loss, and raw emulsion color. Cooking fat loss linearly increased (*p* = 0.02) while cooked emulsion L* linearly decreased (*p* = 0.04) as silver skin level increased. Hardness, gumminess, and chewiness decreased linearly as silver skin levels increased (*p* < 0.01). Overall, incorporating silver skin into meat emulsions reduced stability, increased fat loss, and led to a weaker cooked emulsion matrix.

## 1. Introduction

According to the National Health and Nutrition Examination Survey [1], processed meats account for approximately 21% of all meats consumed in the U.S. Further processing does not only add value to less desirable cuts, but also allows the meat industry to offer products with longer shelf-life, color stability, and unique flavors [2]. According to Tarté [3], cooked processed meats may be obtained from an emulsified meat batter composed of a mixture of finely comminuted meat, fat, salt, and spices. Because of the high-fat content usually found in these emulsion-type products (around 20%), the presence of proteins as emulsifying agents stabilizes the meat batter by creating smaller and well-dispersed fat globules [4]. If the emulsion is not stable enough, excessive water and fat are lost during cooking leading to detrimental effects on yields and on overall product quality [5].

Meat proteins are often used as emulsifier agents in the meat batter due to their amphiphilic structures, however, the myofibrillar ones have the best emulsifying properties [6]. On the other hand, stromal proteins, which are mainly composed of collagen, have limited emulsification capacity [7]. In a study evaluating the stability of pork emulsions using meat sources from different breeds, Sorapukdee et al. [8] showed that the content of stromal proteins was positively associated with water and fat loss, while myofibrillar protein content was associated with higher water-holding capacity and lower fat loss. Previous studies indicated that increasing collagen content in meat emulsions decreased product yields and affected texture profiles. Yong et al. [9] demonstrated that the addition of duck skin, as a source of raw collagen, did not affect total cooking loss, but increased fat loss in cooked pork emulsions. In a different study, Petridis et al. [10] added pork skin as a source of raw collagen in low-fat frankfurters and observed lower product hardness.

However, the inclusion of extracted collagen fiber products as a functional additive in meat emulsions usually improves product yields by increasing water-holding capacity [11]. These additives are obtained by extracting or concentrating collagen from collagen-rich tissues such as skin and bones [3]. These collagen-rich extracts can also be hydrolyzed to improve water solubility [3]. Using a commercially available collagen fiber extracted from bovine hide, Pereira et al. [12] observed a linear decrease in total cooking loss as the level of collagen fiber increased from 0 to 1% in emulsion-type sausages. The yield of cooked emulsion-type sausages also increased with the inclusion of extracted pork collagen [13]. Therefore, the effect of collagen on meat emulsion seems to be dependent on processing and source.

The difference among collagen sources may be associated with fiber type and solubility. For instance, in the skeletal muscle, the endomysium is predominantly formed by collagen fiber types III, IV, and V, while perimysium by types I and III, and epimysium by type I [14]. It has been suggested that type I collagen fiber is more soluble than type III [15], indicating that among these three layers of connective tissues, epimysium would have the greater collagen solubility. The epimysium, also known as silver skin, is a thick fascia of connective tissue that surrounds each muscle, and it can be easily removed from large beef cuts. When fabricating beef sub-primal cuts, silver skin is trimmed and usually discarded. However, since this byproduct is rich in collagen, it is possible to use it as a low-cost ingredient in processed meat formulations to increase product yield and water-holding capacity, as observed for other collagen sources [12,13]. Currently, there is limited scientific evidence about the emulsifying properties of beef silver skin. Thus, the objective of this study was to evaluate the effect of beef silver skin (epimysium) inclusion on meat emulsion stability and the quality of cooked emulsified products.

## 2. Materials and Methods

### 2.1. Ingredient Preparation

Approximately 11 kg of fresh pork loins (*Longissimus* Muscle) were procured from three local supermarkets (source A, B, and C) in Bowling Green, Kentucky, and brought to the Western Kentucky University (WKU) Meat Science lab. All visible fat and connective tissue were removed prior to grinding. Meat was ground once through a 16.6-mm plate, and subsequently, ground twice through a 7.5-mm plate (KG-22-WXP, Pro-Cut, Houston, TX, USA). Ground pork was portioned according to source and treatment formulation (Table 1), placed in Ziploc bags, and stored at −20 °C until the preparation of meat emulsions. Frozen ground beef and beef fat trimmings obtained directly from the WKU Meat Science Laboratory were thawed at 4 °C for 12 h and ground twice through a 7.5-mm plate. A whole batch of silver skin (SS) trimmed from beef tenderloin (*Psoas major*) was obtained directly from a local butcher, sliced, and frozen at −20 °C. The frozen SS was then ground once through a 16.6-mm plate, followed by twice through a 7.5-mm plate. Ground beef, fat trimmings, and SS were portioned according to individual treatment formulations (Table 1), placed in Ziploc bags, and stored at −20 °C until use. A subsample of 200 g of each ingredient was obtained for later chemical composition, as described in Section 2.8. The curing salt and the frankfurter spice mix were obtained from the Great American Spice Company (Rockford, MI, USA), while salt and ice were purchased from a local grocery store. The curing salt, salt, and spice mix were weighed and stored in air-sealed containers at room temperature until emulsions were prepared.

### 2.2. Experimental Design and Emulsion Preparation

The experimental design is shown in Figure 1. Within each block (pork source), three emulsion batches were created, which contained no additional beef silver skin (SS-0), 5% of silver skin (SS-5), or 10% of silver skin (SS-10). The trial was replicated for three consecutive days. In order to maintain protein and fat percentage levels similar, silver skin replaced ground pork, ground beef, and beef fat trimmings (Table 1 and Table 2).

At first, frozen ingredients were thawed at 4 °C for 12 h and mixed in the respective proportions (Table 1) in a bowl chopper (84181D, Hobart, Troy, OH, USA) at 1725 rpm for nine minutes, with the spice mix, salt, curing salt, and crushed ice added after two minutes of mixing. Between batches, the bowl was cleaned and placed at −18 °C for 30 min. Immediately after chopping, the emulsion temperature was recorded using a waterproof digital thermometer (9848EFDA, Taylor Oak Brook, IL, USA). 

### 2.3. pH

The pH of the individual ingredients and meat emulsion was measured by using a portable pH meter (Accumet AE6, Fisher Scientific, Waltham, MA, USA) equipped with a pH/temperature probe (Eutech ECFC7252201B, Fisher Scientific, Waltham, MA, USA). Before use, the pH meter was calibrated using standard buffers at pH 4.01 and 7.00. The pH of meat ingredients was recorded immediately after blending (Classic series 40, Oster, South Shelton, CT, USA) 10 g of sample with 100 mL of distilled water for 30 s. Freshly mixed meat emulsion pH was evaluated by inserting the probe directly into each batch at three points.

### 2.4. Emulsion Stability

The emulsion stability was assessed as described by Sorapukdee et al. [8]. Briefly, in duplicate, 25 g of fresh emulsion was stuffed into 50 mL centrifuge tubes using a flavor injector (Good Grips^®^, OXO, New York, NY, USA). Tubes were capped and cooked in a water bath (Model 186, Precision Scientific Group, Chicago, IL, USA) at 70 °C for 30 min. The tubes with the cooked emulsion were inverted into a glass funnel mounted on a glass culture tube to recover all the cooking fluid loss. The mass of the liquid loss during cooking was recorded, and the culture tubes were dried overnight in an oven with the temperature set at 105 °C. The culture tubes were weighted again to determine the fat loss. The total loss, water loss, and fat loss are shown as a percentage of the fresh sample. All samples were analyzed in duplicates.

### 2.5. Instrumental Colors

The instrumental color was evaluated using a MiniScan EZ colorimeter (4500 L, Hunter Lab, Reston, VA, USA), which was calibrated according to the manufacturer’s recommendations before readings. The equipment was set to use the illuminator D65, an angle of 10° to the observer. Values of lightness (*L**), redness (*a**), and yellowness (*b**) were obtained according to CIE *L*a*b** color space. For raw emulsions, color was assessed on 5 different points on the surface of the batter that was evenly distributed in a plastic petri dish. For cooked products, color was measured one day after the emulsion preparation on the same samples used for emulsion stability. The cooked emulsion was diced, placed into a plastic petri dish, and five measurements were performed across the sample surface.

### 2.6. Water Activity

The water activity (a_w_) of each emulsion was measured in duplicates by using an AquaLab water activity meter (Model CX-2, Decagon Devices Inc., Pullman, WA, USA). The instrument was previously calibrated using the following standards 0.250, 0.500, 0.760, 0.920, 0.984, and 1.000.

### 2.7. Collagen Analysis

Collagen content was calculated from colorimetric measurement of hydroxyproline after acid hydrolysis [16,17]. In duplicate, approximately 2 g of each ingredient and emulsion samples were placed in an Erlenmeyer flask with 25 mL of 6 M HCl. Flasks were fitted with a rubber stopper and secured with wire to prevent the stopper from ejecting during digestion. The flasks were placed in an oven at 105 °C for 12 h. Hydrolyzed samples were filtered through #2 filter paper, and the pH was adjusted to the range between 6.5 and 7 by adding 10 M NaOH and distilled water. The filtrate was then transferred into a volumetric flask and the volume was brought up to 250 mL with distilled water. The solution was then used to quantify hydroxyproline using a commercial kit (Hyp Assay Kit abx298833, Abbexa LLC, Houston, TX, USA). Collagen concentration was then obtained by multiplying hydroxyproline concentration by 7.143, considering that meat collagen contains approximately 14% hydroxyproline [18].

### 2.8. Chemical Composition

Approximately 200 g of each ingredient and meat emulsions were dried in a forced air oven at 60 °C for 12 h. Dried samples were frozen at −18 °C and then ground using a mortar and pestle to pass through a 1 mm screen. Grinding was performed in a cooling room at 5 °C. An aliquot of 10 g was sent to Dairy One forage lab (Ithaca, NY, USA) for moisture, fat, and protein analyses. Moisture content was determined by oven drying samples at 105 °C for 3 h (NFTA Method 2.2.2.5). Total fat was analyzed by acid hydrolysis followed by solvent extraction (AOAC Method 954.02). Briefly, approximately 750 mg of the sample was weighed into an XT4 filter bag (Ankom Technology Inc., Macedon, NY, USA) containing 750 mg of diatomaceous earth. The bag was sealed and hydrolyzed with 4 N hydrochloric acid for 60 min in a sealed Teflon vessel (ANKOM^HCl^ Hydrolysis System, Ankom Technology Inc., Macedon, NY, USA). After hydrolysis, the fat extraction was performed using an ANKOM XT15 Extractor with solvent solution (45% petroleum ether, 45% diethyl ether, and 10% ethanol at 90 °C for 60 min. Nitrogen content was analyzed by complete combustion (AOAC Method 992.15) using a carbon/nitrogen determinator (CN628, LECO Corporation, St Joseph, MI, USA). Crude protein was then obtained by multiplying nitrogen content by 6.25.

### 2.9. Texture Profile Analysis

In duplicate, approximately 25 g of freshly mixed emulsion was stuffed into 50 mL centrifuge tubes, using a flavor injector (Good Grips^®^, OXO, New York, NY, USA). The tubes were capped and cooked in a water bath at 70 °C for 30 min. The samples were randomly labeled, refrigerated overnight at 5 °C, and shipped to the Nevada Meat Science Laboratory at the University of Nevada, Reno, for texture profile analysis (TPA). The cooked samples with a cylindric shape (30 mm in diameter and 15 mm in length) were evaluated for TPA following the double compression test. Samples were compressed to 50% of their diameter using a computer-controlled texture analyzer (TMS-PRO, Food Technology Corporation, Sterling, VA, USA) fitted with a 500 N load cell and a 65 mm compression probe. The crosshead speed used was 200 mm/min and the trigger force was 5 N. The following parameters were assessed: hardness, adhesiveness, cohesiveness, springiness, gumminess, and chewiness [19].

### 2.10. Statistical Analysis

Data were first checked for outliers (>2.5 Studentized Residual) and then for normality by Shapiro-Wilk test using PROC UNIVARIATE of SAS (version 9.4; SAS Institute Inc., Cary, NC). The ANOVA was performed using the PROC MIXED of SAS including the fixed effect of treatment (silver skin level), the random effect of block (pork source; *n* = 3), and repeated measures over time (day; *n* = 3). Five covariance structures of the R matrix were tested (autoregressive, compound symmetry, unstructured, Toeplitz, and variance component), and the one yielding the lowest Bayesian information criterion (BIC) was kept in the final model. The order of mixing was tested as a covariate and kept in the model when *p* ≤ 0.10. Orthogonal polynomials were used to test linear and quadratic responses of silver skin inclusion on emulsion quality traits. All reported values are least squares means (LSM) ± standard error of the mean (SEM). Significance was declared at *p* ≤ 0.05 and trends at 0.05 < *p* ≤ 0.10. The PROC CORR of SAS was used to compute the Pearson partial correlation coefficients among the independent variables adjusted for silver skin level, pork source, and day of production.

## 3. Results and Discussion

### 3.1. Chemical Composition and Collagen Concentration

Table 3 shows the chemical composition of the raw meat emulsions. As the silver skin level increased, no significant changes in emulsion composition were observed (*p* > 0.05). Total collagen concentration increased linearly (*p* = 0.008) in response to the increase in silver skin level. These results demonstrated that including silver skin at the expense of fat trimmings, ground beef, and ground pork, increased collagen concentration without affecting the chemical composition of the meat emulsions, which reduces the risk of potential confounding effects since fat and protein levels can affect emulsion characteristics [5,20,21].

The fat percentage averaged 19.6%, which is lower than that found in commercially available emulsion-type sausages (i.e., frankfurter) in the U.S., where fat levels range from 25–31% of fat [22], and it is below the limit of 30% allowed by U.S. regulations [23]. Meat emulsion moisture and protein averaged 65% and 12.4% respectively, which was higher than the 54.6% moisture but similar to the 11.7% protein reported in the national survey for emulsion-type sausage [22].

### 3.2. Physical Parameter and Emulsion Stability

The physical parameters of the raw emulsion are presented in Table 4. Silver skin inclusion tended (*p* = 0.079) to increase linearly raw emulsion pH. This is likely due to the higher pH of silver skin compared to the other ingredients (Table 2). Although emulsion pH can affect meat protein interaction with water [24], the pH change observed here was only 0.04 from the level 0 to 10% of silver skin (5.68 vs. 5.72) and most likely did not affect any other parameter such as water-holding capacity. This assumption can be confirmed by the lack of correlation between pH and other emulsion traits evaluated (Figure 2).

The inclusion of silver skin did not affect water activity (*p* > 0.05), which averaged 0.986. Similar results were reported by Pereira et al. [12], who found no effect of the inclusion of collagen fiber extracted from bovine hide on water activity in frankfurter-type sausages. Meat emulsion temperature linearly increased with the inclusion of silver skin (*p* = 0.019). An increase in emulsion temperature was also observed by Jones et al. [25] when modified connective tissue from beef was included in meat batters. The increment in temperature may be an effect associated with the friction created while chopping the coarser structure of collagen [26]. Temperature control is a critical parameter to be monitored when mixing the meat batter since overheating might cause protein denaturation, decreasing the emulsion stabilizing ability attributed to meat proteins. Bañón et al. [20] found that cooking loss increases rapidly as emulsion temperature increases from 0 to 50 °C. Therefore, it has been suggested that meat emulsion temperature should not rise above 10 °C during chopping to prevent detrimental effects on water-holding capacities [27].

Cooking and water losses were not affected by beef silver skin inclusion (*p* > 0.05), suggesting that cooking yield was not affected by collagen level. However, fat loss linearly increased with the inclusion of silver skin (*p* = 0.026). The lack of differences in water loss when comparing samples from different treatments can be explained by the decrease in surface hydrophobicity of type I collagen after cooking at 70 °C for 30 min [28]. Similar results were found by Yong et al. [9], when the inclusion of duck skin, a rich source of raw collagen, did not affect total cooking loss but increased fat loss. In contrast, cooking loss decreased linearly as extracted collagen fiber (from bovine hide) levels increased in frankfurter-type sausages [12]. In another study, the yield of cooked frankfurters increased with the inclusion of 1% of a commercial pork collagen product [13]. These results show that the nature of the collagen (i.e., extracted vs. raw) affects its ability to bind water. The greater fat loss in response to silver skin inclusion can be a consequence of the low fat-holding capacity of collagen, as previously described [5]. A positive correlation (*r* = 0.76) between fat released and stromal protein concentration has been reported for cooked pork emulsion [8]. In our study, the correlation between collagen concentration and fat loss did not differ from zero (*r* = 0.10, *p* = 0.66). This result suggests that the increase in fat loss, in response to silver skin inclusion, may be linked to other factors such as emulsion temperature, which showed a positive correlation with fat loss (*r* = 0.49, *p* = 0.02).

The instrumental color of the raw emulsion was not affected by beef silver skin level (*p* > 0.05). This lack of difference in raw emulsion color can be explained by the similar chemical composition of meat emulsions with different levels of silver skin (Table 3). Previous studies have shown that meat emulsions formulated with different chemical compositions present a distinct color [5,20]. Cooked emulsion redness (*a**) and yellowness (*b**) were not affected by beef silver skin level (*p* > 0.05). On the other hand, cooked emulsion lightness (*L**) decreased as beef silver skin level increased (*p* = 0.045). This reduction in cooked emulsion lightness was most likely caused by the increased fat loss, which led to lower light reflection. Lower *L** has been reported for meat emulsions with reduced fat content, demonstrating the influence of fat content on emulsion lightness [20,21].

### 3.3. Texture Profile Analysis

The results of the texture profile analysis are presented in Table 5. Silver skin inclusion affected all texture parameters (*p* < 0.05). Cooked emulsion hardness linearly decreased as beef silver skin inclusion increased (*p* = 0.001), suggesting that adding beef silver skin led to a weaker cooked emulsion structure. In addition, the collagen concentration in the meat emulsion was negatively correlated with hardness (*r* = −0.61, *p* = 0.04). In a study evaluating two inclusion levels of pork skin (13 and 27%) as a source of non-hydrolyzed collagen in low-fat frankfurters (15% of lard), Petridis et al. [10] also found a negative association between pork skin inclusion and hardness. In contrast, greater hardness of emulsion-type products was observed by adding hydrolyzed porcine collagen [29] or collagen fiber extracted from bovine hide [12]. These conflicting results are most likely caused by the distinct ability of collagen sources to stabilize the emulsion. Extracted and hydrolyzed collagen sources are more soluble and have greater gel-forming capacity than the native collagen fibers in raw meat products, such as silver skin [3]. Therefore, due to the lower emulsion stability, larger fat droplets are formed which are more susceptible to leakage during the cooking process [5]. This assumption is supported by a previous study where greater hardness was observed in treatments with higher fat loss [30].

Adhesiveness and springiness of cooked emulsion were linearly (*p* < 0.001) and quadratically (*p* ≤ 0.001) affected by beef silver skin inclusion, showing that there was a greater difference between the treatments SS0 and SS5 than between the SS5 and SS10. Similarly, cohesiveness was quadratically affected by beef silver skin inclusion (*p* = 0.02) with a higher value observed for SS5 compared to SS0 and SS10. These results indicate that adding beef silver skin made the cooked emulsion less sticky (lower adhesiveness) and less elastic (lower springiness) under compression. In line with our findings, Petridis et al. [10] reported a positive relationship between cohesiveness and the inclusion of raw pork skin as a source of collagen. In contrast, Sousa et al. [29] found no change in adhesiveness, cohesiveness, and springiness when hydrolyzed pork collagen was added to frankfurter sausages. Cohesiveness and adhesiveness were also not affected by adding extracted collagen fiber to frankfurter-type sausages [12]. These results evidenced that raw collagen sources affect more adhesiveness, cohesiveness, and springiness than extracted or hydrolyzed sources.

Gumminess and chewiness were linearly decreased as beef silver skin inclusion increased (*p* < 0.001). This result agrees with Sorapukdee et al. [8], who found a negative correlation between stromal protein and either gumminess or chewiness. In the current study, collagen concentration in the emulsion was negatively correlated with gumminess (*r* = −0.53, *p* = 0.02) and chewiness (*r* = −0.45, *p* = 0.05). In a study evaluating two inclusion levels of pork skin (13 and 27%) as a source of non-hydrolyzed collagen in low-fat frankfurters (15% of lard), Petridis et al. [10] also observed a negative association between pork skin inclusion and gumminess. Additionally, these authors observed that consumers preferred the frankfurters with 13% pork skin, indicating that the reduction in hardness and gumminess had a negative impact on sensorial attributes [10]. Altogether, our data show that the inclusion of silver skin as a source of raw collagen seems to create a weaker meat emulsion matrix leading to a decrease in the texture profile parameters.

## 4. Conclusions

Although the inclusion of beef silver skin may decrease production costs, the results of this research suggested that silver skin may reduce emulsion stability. The effect of the inclusion of beef silver skin was more detrimental to fat-holding capacity than water-holding capacity. Consequently, silver skin inclusion promoted fat loss during cooking and reduced emulsion lightness. All the texture profile parameters were negatively affected by silver skin inclusion, which is most likely a consequence of a weaker meat emulsion matrix. Further studies should be implemented to investigate if these detrimental impacts on meat product quality could be avoided by extracting or hydrolyzing collagen from silver skin instead of using it as a raw source. As an implication, sausage makers should be aware that an elevated level of beef silver skin (epimysium) trimmings can impair emulsion-type product quality.

## Figures and Tables

**Figure 1 foods-12-03775-f001:**
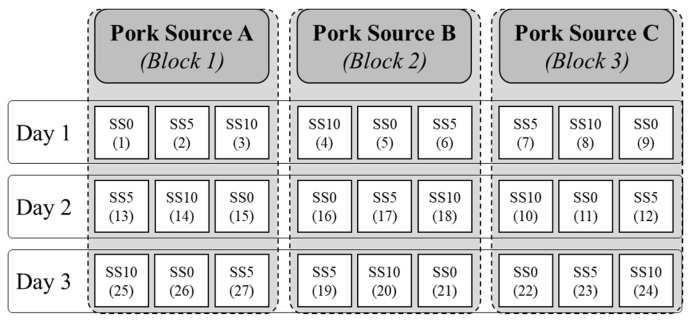
The experimental design used to evaluate the beef silver skin (SS) inclusion levels (at 0, 5, and 10%) on meat emulsion and product quality. Numbers in parentheses represent the batch order.

**Figure 2 foods-12-03775-f002:**
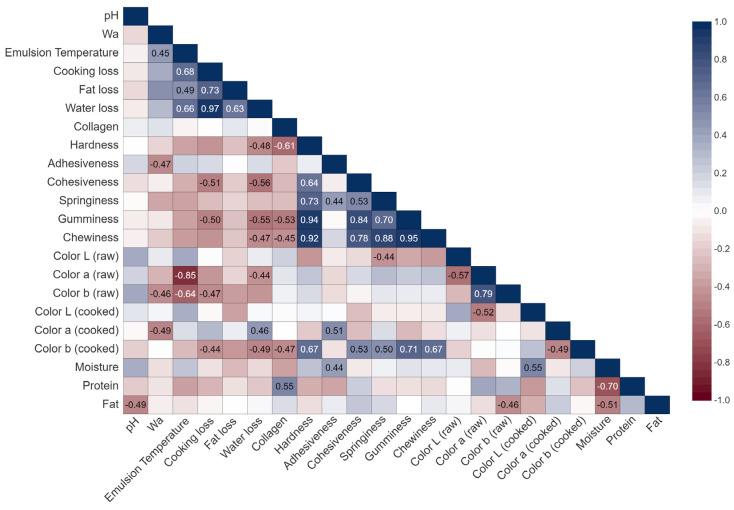
Heatmap showing the Pearson partial correlation between the evaluated variables on raw and cooked emulsion. The correlation coefficients shown are significantly different than zero *p* ≤ 0.05.

**Table 1 foods-12-03775-t001:** Emulsion formulations according to the inclusion level of beef silver skin (SS).

Ingredients (g)	Treatments		
	SS-0	SS-5	SS-10
Ground pork	366	344	322
Ground beef	184	172	160
Fat trimming	184	172	160
Crushed ice	184	184	184
Silver skin	0	46	92
Cure salt ^1^	1.03	1.03	1.03
Salt (sodium chloride)	9.2	9.2	9.2
Spice mix ^2^	14	14	14

^1^ Contains 6.25% sodium nitrite and 93.75% sodium chloride. ^2^ Frankfurter & Bologna Complete Mix from Great American Spice.

**Table 2 foods-12-03775-t002:** Moisture, protein, fat, collagen, and pH of the ingredients used.

Item	Ground Beef	Silver Skin	Pork Source A	Pork Source B	Pork Source C	Beef Fat Trimmings
Moisture, %	56.7	47.9	75.5	72.5	71.9	20.5
Protein, %	15.4	15.2	20.8	20.7	22.4	1.59
Fat, %	22.8	35.6	1.37	3.13	1.68	77.8
Collagen, mg/g	16.9	53.9	5.77	6.68	7.28	12.1
pH	5.86	5.96	5.85	5.71	5.80	5.89

**Table 3 foods-12-03775-t003:** Composition, and total collagen of the emulsion products as affected by beef silver skin inclusion level.

	Silver Skin Level ^1^				*p*-Value	
Item	0	5	10	SEM ^2^	Linear	Quadratic
Moisture, %	65.0	64.9	65.1	0.349	0.880	0.679
Protein, %	12.4	12.4	12.3	0.088	0.497	0.140
Fat, %	19.4	19.8	19.2	0.512	0.856	0.389
Total collagen, mg/g	7.66	8.71	9.32	0.392	0.008	0.676

^1^ Beef silver skin inclusion at the levels of 0, 5, and 10%. ^2^ SEM = Standard error of the mean.

**Table 4 foods-12-03775-t004:** Physical parameters (pH, water activity, temperature, and color) of raw emulsion, cooking losses, and cooked emulsion color as affected by silver skin inclusion level.

	Silver Skin Level ^1^				*p*-Value	
Item	0	5	10	SEM ^2^	Linear	Quadratic
pH	5.68	5.69	5.72	0.014	0.079	0.828
Water activity	0.984	0.987	0.988	0.002	0.133	0.623
Emulsion temperature, °C	7.57	8.93	8.96	0.60	0.019	0.165
Total cook loss, %	12.8	13.0	15.2	1.41	0.136	0.477
Water loss, %	11.5	11.6	12.9	1.17	0.225	0.584
Fat loss, %	1.39	1.92	2.27	0.28	0.026	0.770
Raw emulsion color						
* L**	63.4	64.0	63.8	0.72	0.671	0.553
* a**	6.18	5.88	6.11	0.15	0.769	0.165
* b**	17.7	17.5	17.6	0.16	0.746	0.302
Cooked emulsion color						
* L**	64.1	63.7	62.9	0.38	0.045	0.679
* a**	7.80	7.57	8.04	0.28	0.368	0.132
* b**	13.2	13.5	13.7	0.31	0.314	0.967

^1^ Beef silver skin inclusion at the levels of 0, 5, and 10%. ^2^ SEM = Standard error of the mean.

**Table 5 foods-12-03775-t005:** Texture profile of cooked emulsion as affected by silver skin inclusion level.

	Silver Skin Level ^1^				*p*-Value	
Item	0	5	10	SEM ^2^	Linear	Quadratic
Hardness, N	36.9	33.8	32.9	1.03	0.001	0.244
Adhesiveness, N.mm	0.873	0.598	0.641	0.050	<0.001	<0.001
Cohesiveness, ratio	0.240	0.248	0.232	0.009	0.171	0.024
Springiness, ratio	3.14	2.80	2.78	0.065	<0.001	0.001
Gumminess, N	8.78	8.44	7.70	0.439	<0.001	0.321
Chewiness, N	27.9	24.2	22.1	1.97	<0.001	0.437

^1^ Beef silver skin inclusion at the levels of 0, 5, and 10%. ^2^ SEM = Standard error of the mean.

## Data Availability

The data used to support the findings of this study can be made available by the corresponding author upon request.

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
