# Peer review of "Effect of Beef Silver Skin (Epimysium) Levels on Meat Emulsion Stability, Quality Attributes, and Texture Parameters"

_foods, 2023, doi:10.3390/foods12203775_

Round 1
Reviewer 1 Report (Previous Reviewer 3)
Comments and Suggestions for Authors
The manuscript by Kawata et al is now improved and may be accepted for publication. Authors have edited it to the maximum possible level and included the issue of cost and emulsifying capacity. The abstract and methodology now clear and informative.
Author Response
Thank you for your effort in helping us improve our manuscript!
Reviewer 2 Report (New Reviewer)
Comments and Suggestions for Authors
The article presents the rationale for the use of small additions of silver skin (5 or 10%) to raw meat, which do not lead to a deterioration in its meat emulsion stability and culinary properties. The article is of great technological interest for workers in the meat industry, given its large size, as well as for all readers of the Foods.
I think that the article can be improved taking into account the following comments and suggestions:
-Table 3: From this table it follows that adding 5% of silver skin leads to an increase in collagen by 8.97-7.5 = 1.47 mg/g, while adding another 5% leads to an increase in collagen only by 9.44-8.97 = 0.47 mg/g. The difference is about three times. In my opinion, there is some kind of mistake here.
- Line 135: “lightness (L*), redness (a*), and yellowness (b*)”. Not everyone knows what the asterisks here mean. Can you give the designations without asterisks?
- Line 262: “A significant and positive correlation (r = 0.76)”. Correlation 0.99 is really significant, but in this case it is simply positive.
Author Response
The article presents the rationale for the use of small additions of silver skin (5 or 10%) to raw meat, which do not lead to a deterioration in its meat emulsion stability and culinary properties. The article is of great technological interest for workers in the meat industry, given its large size, as well as for all readers of the Foods. I think that the article can be improved taking into account the following comments and suggestions:
-Table 3: From this table it follows that adding 5% of silver skin leads to an increase in collagen by 8.97-7.5 = 1.47 mg/g, while adding another 5% leads to an increase in collagen only by 9.44-8.97 = 0.47 mg/g. The difference is about three times. In my opinion, there is some kind of mistake here.
AU. We agree that the difference should be more consistent among treatments since they are equally spaced. Therefore, we double-checked all the raw data and there was no wrong calculation or outliers (based on Studentized Residual greater than ±2.5). However, after running Influence Diagnostics (option influence on PROC MIXED), we detected an influential observation (based on Cook’s D value). We removed the data point and ran the statistical analysis again, which now best fitted the TOEP covariance structure, which affected all the LSmeans, SEM, and of course P-values. Treatments LSmeans are still not equally spaced, but the quadratic effect is now non-significative (please see Table 3). Thanks for suggesting that we review this data!
- Line 135: “lightness (L*), redness (a*), and yellowness (b*)”. Not everyone knows what the asterisks here mean. Can you give the designations without asterisks?
AU. Based on the AMSA guidelines for meat color measurement (you can find at:
https://meatscience.org/publications-resources/printed-publications/amsa-meat-color-measurement-guidelines), the presence or absence of the asterisk reflects the mathematical method used by the equipment. Since we used the MiniScan EZ colorimeter model 4500L (please see L.132), this equipment gives us the data according to the CIEL*a*b* color space. Therefore, the correct way to report it is with the asterisk.
- Line 262: “A significant and positive correlation (r = 0.76)”. Correlation 0.99 is really significant, but in this case it is simply positive.
AU. We changed this sentence to “A positive correlation” (now L.263). Thank you!
Reviewer 3 Report (New Reviewer)
Comments and Suggestions for Authors
Manuscript foods-2670059, entitled “Effect of beef silver skin (epimysium) levels on meat emulsion stability, quality attributes, and texture parameters”
Recommendation: The above paper needs to a major modify.
This article provides useful information on the effects of different beef epimysium levels on meat emulsion stability, quality attributes, and texture parameters. It is in general appropriately organized, carried out and written, however there are some points that should be corrected or clarified.
L69: The statement here is contrast with that in L46-47
L69: “observed” instead of “seen”
Figure 1: You refer to “pork source”, however beef and fat were also added at a high percentage
L101: “At first, frozen…”
Table 2: Is it expected that SS-10 has a lower fat level compared to SS-5 comparable to that of SS-0?
L139: “…and five measurements were performed across…”
L158: “Approximately 200 g of each…”
L193-194: “All values are reported as least squares means (LSM) ± standard…”
L198: Normally the parts of “Results” and “Discussion” are separate
L201-202: “…observed (P > 0.05). Total collagen concentration increased linearly (P<0.001) and quadratically (P = 0.042) in response to the increase of silver skin level. This effect of…”
L207: Please rephrase “which avoid any confounding”
L218: 55 and 65% are similar?
L221: “The physical parameters of the raw emulsion are presented…”
Table 3, 4 and 5: Please also provide ANOVA P-value
L262: “described” instead of “suggested”
L282: “…in the meat emulsion was negatively…”
L299: Adhesiveness and springiness were also linearly affected
L303-304: “In line with our findings, Petridis et al. [10] reported a…”
L310-311: “Gumminess and chewiness were linearly decreased as beef silver skin inclusion increased (P < 0.001). This result agrees with that of Sorapukdee et al. [8], who…”
L327: “implemented” instead of “done”
Comments on the Quality of English Language
Minor editing of English language required
Author Response
This article provides useful information on the effects of different beef epimysium levels on meat emulsion stability, quality attributes, and texture parameters. It is in general appropriately organized, carried out and written, however there are some points that should be corrected or clarified.
L69: The statement here is contrast with that in L46-47
AU. In L69, we are referring specifically to beef silver skin, while L46 is collagen from purified sources. As shown throughout the introduction section, there are conflicting results in response to collagen concentration, depending on the collagen source.
L69: “observed” instead of “seen”
AU. We changed the word accordingly. Thank you!
Figure 1: You refer to “pork source”, however beef and fat were also added at a high percentage
AU. Ground pork was our main ingredient (please see Table 1), and to create true replicates we purchased pork from three sources (L.75-76). The other ingredients were obtained from a single source. Pork source was included as a block effect in the statistical model.
L101: “At first, frozen…”
AU. We changed the word accordingly. Thank you!
Table 2: Is it expected that SS-10 has a lower fat level compared to SS-5 comparable to that of SS-0?
AU. Are you referring to Table 3? We were not expecting differences in fat content because as SS was included, lean beef, pork, and fat trimmings were replaced to obtain a similar chemical composition (L.95-97 and Table 3).
L139: “…and five measurements were performed across…”
AU. We changed the word accordingly. Thank you!
L158: “Approximately 200 g of each…”
AU. We deleted the “,” (now L.159)
L193-194: “All values are reported as least squares means (LSM) ± standard…”
AU. We added “±” (now L.195)
L198: Normally the parts of “Results” and “Discussion” are separate
AU. According to FOODS instructions for the authors; “The results section may be combined with Results.” As authors, we decided to combine to create a concise and informative paper. Thanks for your comments.
L201-202: “…observed (P > 0.05). Total collagen concentration increased linearly (P<0.001) and quadratically (P = 0.042) in response to the increase of silver skin level. This effect of…”
AU. We have made these changes accordingly (now L. 202-203). Please, note that LSmeans for total collagen slightly changed after reviewing our data following the suggestion of another reviewer (see also Table 3).
L207: Please rephrase “which avoid any confounding”
AU. We have rephrased this sentence (now L. 206).
L218: 55 and 65% are similar?
AU. We agree it is not similar. We rephrased this sentence (now L.217-218). Thank you for identifying and bringing this error to my attention.
L221: “The physical parameters of the raw emulsion are presented…”
AU. We have made these changes accordingly (now L.220).
Table 3, 4 and 5: Please also provide ANOVA P-value
AU. We respectfully disagree with this suggestion. Although we already have the P-values for the fixed effect (SS inclusion), including it would not be informative because we don’t have only two treatments. Therefore, the treatment P-value of treatment doesn’t stand alone and can not be used for any interpretation.
L262: “described” instead of “suggested”
AU. The word was changed accordingly (now L. 261).
L282: “…in the meat emulsion was negatively…”
AU. We inserted the word “was” (now L. 281).
L299: Adhesiveness and springiness were also linearly affected
AU. Thanks for your suggestion. We modified this sentence accordingly (now L.300-302).
L303-304: “In line with our findings, Petridis et al. [10] reported a…”
AU. We added the direct citation, as suggested (now L. 304).
L310-311: “Gumminess and chewiness were linearly decreased as beef silver skin inclusion increased (P < 0.001). This result agrees with that of Sorapukdee et al. [8], who…”
AU. We added “were” to the sentence.
L327: “implemented” instead of “done”
AU. We replaced this word, as suggested. Thank you!
This manuscript is a resubmission of an earlier submission. The following is a list of the peer review reports and author responses from that submission.
Round 1
Reviewer 1 Report
Comments and Suggestions for Authors
A paper Effect of beef silver skin (epimysium) levels on meat emulsion stability, quality attributes, and texture parameters is interesting with all supporting data.
Few minor things are necessary to correct:
-in Materials and methods correct N for M, it's the right way for using molarity
-in line 180 put space 200 mm/min
Author Response
Reviewer #1
paper Effect of beef silver skin (epimysium) levels on meat emulsion stability, quality attributes, and texture parameters is interesting with all supporting data.
Few minor things are necessary to correct:
-in Materials and methods correct N for M, it's the right way for using molarity
AU. We have made the suggested changes (L. 147 and L. 150).
-in line 180 put space 200 mm/min
AU. Thanks for catching that mistake.
Reviewer 2 Report
Comments and Suggestions for Authors
This study suffers from various shortfalls with respect to the problem statement, hypothesis, experimental design, and presentation of results. It is stated in the introduction section that there is limited scientific evidence about the emulsifying properties of beef silver skin. It is exactly for the same reason that this study should have focused on mapping the emulsification potential of beef silver skin before studying its functionality directly in the product matrix. For example, the authors should have analyzed the critical micelle concentration and its emulsion formation and stabilization indices.
- How did the authors arrive at the concentration of 5% and 10% of beef silver skin? Is there any preliminary data based on which the authors narrowed down at these levels? From the formulation, it seems that the beef silver skin has been added at 5% and 10% of the total weight of the other ingredients rather than as a percentage of the fat and water, the interface of which the emulsifier is supposed to stabilize. Can the authors justify this?
- The values of standard deviation are missing in Tables 2-5.
- What is the statistical significance of the difference in pH between beef silver skin and other ingredients? The same has been stated as the reason for the increase in pH of the raw emulsion with the inclusion of beef silver skin.
- What is the classification of collagen in the beef silver skin used in this study? (Type-1/Type-2?)
- When it is already known that stromal proteins that are mainly composed of collagen have limited emulsification capacity, increasing collagen content in meat emulsions decreased product yields and affected texture profile, and inclusion of extracted collagen fiber products as a functional additive in meat emulsions improves product yields by increasing water-holding capacity (as mentioned in the introduction), why the authors have not considered all these knowhows while designing this study? Did the authors investigate how to modify or functionalize the beef silver skin to render it a suitable emulsifier for emulsion-type meat products?
- Though the authors have mentioned that their data showed that the inclusion of silver skin as a source of raw collagen created a weaker meat emulsion matrix with reduced texture profile parameters, they should have also worked on a solution to this based on their observations and published works (for ex., hydrolysis).
Comments on the Quality of English Language
The language of the manuscript is appropriate.
Author Response
Reviewer #2
This study suffers from various shortfalls with respect to the problem statement, hypothesis, experimental design, and presentation of results. It is stated in the introduction section that there is limited scientific evidence about the emulsifying properties of beef silver skin. It is exactly for the same reason that this study should have focused on mapping the emulsification potential of beef silver skin before studying its functionality directly in the product matrix. For example, the authors should have analyzed the critical micelle concentration and its emulsion formation and stabilization indices.
AU. We agree that this study lacks some micro-level analyses that could improve our understanding of this source of collagen in protein-protein and protein-fat interactions. However, our focus here was not basic science but applied meat science with the aim of understanding the impact on product quality and yield, two important factors for the industry.
- How did the authors arrive at the concentration of 5% and 10% of beef silver skin? Is there any preliminary data based on which the authors narrowed down at these levels? From the formulation, it seems that the beef silver skin has been added at 5% and 10% of the total weight of the other ingredients rather than as a percentage of the fat and water, the interface of which the emulsifier is supposed to stabilize. Can the authors justify this?
AU. Due to the lack of research on beef silver skin in emulsion products, we decided to use these three levels based on the availability of the product obtained from a local butcher shop. The highest level of 10% appears to be too high, but we have to consider that this is a flesh trim which consequently includes also small portions of lean and fat tissue (please, see chemical composition in Table 2).
- The values of standard deviation are missing in Tables 2-5.
AU. Table 2 does not have an SD because the chemical analysis was performed in a single subsample from each ingredient. Tables 3-5 have SEM as a measure of statistical dispersion instead of SD.
- What is the statistical significance of the difference in pH between beef silver skin and other ingredients? The same has been stated as the reason for the increase in pH of the raw emulsion with the inclusion of beef silver skin.
AU. We are not able to run a statistical analysis of the pH or any other data in Table 2 because it was measured in duplicate of a single subsample, which means we have no true replicate to run a statistical analysis. The data presented in Table 2 are only to show the chemical composition of the used ingredients, and as we can see silver skin has the highest measure of pH.
- What is the classification of collagen in the beef silver skin used in this study? (Type-1/Type-2?)
AU. Unfortunately, we did not evaluate collagen fiber type in this study. However, according to Sims & Bailey (1981), the epimysium is predominantly composed of type I collagen fibers, as stated in our introduction (L. 59-63).
- When it is already known that stromal proteins that are mainly composed of collagen have limited emulsification capacity, increasing collagen content in meat emulsions decreased product yields and affected texture profile, and inclusion of extracted collagen fiber products as a functional additive in meat emulsions improves product yields by increasing water-holding capacity (as mentioned in the introduction), why the authors have not considered all these knowhows while designing this study? Did the authors investigate how to modify or functionalize the beef silver skin to render it a suitable emulsifier for emulsion-type meat products?
AU. Although we know that in general collagen has limited emulsifying capacity, we don’t know much about this specific collagen source. In addition, data from this study brings some practical applications by showing that silver skin affects product yield and product quality. Therefore, if sausage makers do not trim the fresh meat to decrease silver skin, their product yield and quality can be compromised. This observation would not be possible if the silver skin was processed before its use.
- Though the authors have mentioned that their data showed that the inclusion of silver skin as a source of raw collagen created a weaker meat emulsion matrix with reduced texture profile parameters, they should have also worked on a solution to this based on their observations and published works (for ex., hydrolysis).
AU. We agree that a solution must be generated via research in order to efficiently utilize the by-product without impairing product yield and quality. However, this was out of the scope of the current study, which was a pilot project to highlight the impact of the presence of silver skin in meat emulsion. Our laboratory will continue researching this issue to check the effect of treatments such as hydrolysis and extraction on emulsion stability. Thank you for your comments!
Reviewer 3 Report
Comments and Suggestions for Authors
The manuscript by Katawa et al. reported poor emulsion stability and quality of meat emulsion upon incorporating the epimysium as a collagen source. Authors reported lower emulsion stability and fat loss on the incorporation of beef epimysium. The language is easy and clear. The hypothesis is sound.
I have following observation in this regard-
i. In Abstract: please mention whether the beef epimysium was added by replacing meat? Or other binders/extenders to get better clarity on the subject.
ii. Keywords: Appropriate
iii. Introduction: Appropriate and well-outlined the need for taking the study. Regarding reducing the price, I would also recommend adding this aspect in the abstract.
iv. Methodology: This section, also details the procedure and as per the stated objectives, but I would think better clarity on some aspects would improve the quality and readership of the manuscript as-
During the preparation of meat emulsion, a mixture of pork and beef was used. Please check whether such type of product would not be considered under meat adulteration, or substitution.
Further please add more clarity on the replacement of epimysium, how much replaced with fat and minced meat. Or whether this relative ratio was different for the different formulations.
v. Results and discussion: Appropriate and well supported by the relevant references
Thank you for the opportunity to read your work.
Author Response
Reviewer #3
The manuscript by Katawa et al. reported poor emulsion stability and quality of meat emulsion upon incorporating the epimysium as a collagen source. Authors reported lower emulsion stability and fat loss on the incorporation of beef epimysium. The language is easy and clear. The hypothesis is sound.
I have following observation in this regard-
i. In Abstract: please mention whether the beef epimysium was added by replacing meat? Or other binders/extenders to get better clarity on the subject.
AU. We added the suggested information to the abstract (L.19-20).
ii. Keywords: Appropriate
AU. Thanks for reviewing them.
iii. Introduction: Appropriate and well-outlined the need for taking the study. Regarding reducing the price, I would also recommend adding this aspect in the abstract.
AU. We added this information to the abstract (L.16).
iv. Methodology: This section, also details the procedure and as per the stated objectives, but I would think better clarity on some aspects would improve the quality and readership of the manuscript as-
During the preparation of meat emulsion, a mixture of pork and beef was used. Please check whether such type of product would not be considered under meat adulteration, or substitution.
AU. We understand that in some countries it may be considered an adulteration, but in the U.S. many emulsified products have beef and pork as their ingredients.
Further please add more clarity on the replacement of epimysium, how much replaced with fat and minced meat. Or whether this relative ratio was different for the different formulations.
AU. This information is in the manuscript in the abstract (L.19-20), Material and Methods section (L. 94-96), and in Table 1.
v. Results and discussion: Appropriate and well supported by the relevant references
AU. Thanks for reviewing it!
Round 2
Reviewer 2 Report
Comments and Suggestions for Authors
The explanations provided by the Authors in response to the queries of the initial review are not convincing.
Comments on the Quality of English Language
Minor editing of English language is required.